# Systemic GLP-1R agonist treatment reverses mouse glial and neurovascular cell transcriptomic aging signatures in a genome-wide manner

Zhongqi Li [1,2,10], Xinyi Chen[1,2,10], Joaquim S. L. Vong [2,3,10], Lei Zhao[1,2,10], Junzhe Huang[1,2], Leo Y. C. Yan[1,2], Bonaventure Ip[1], Yun Kwok Wing [4,5,6], Hei-Ming Lai[1,2,4,5,6], Vincent C. T. Mok [1,5,6,7 ✉] & Ho Ko [1,2,4,5,6,7,8,9 ✉]

Pharmacological reversal of brain aging is a long-sought yet challenging strategy for the prevention and treatment of age-related neurodegeneration, due to the diverse cell types and complex cellular pathways impacted by the aging process. Here, we report the genome-wide reversal of transcriptomic aging signatures in multiple major brain cell types, including glial and mural cells, by systemic glucagon-like peptide-1 receptor (GLP-1R) agonist (GLP-1RA) treatment. The age-related expression changes reversed by GLP-1RA encompass both shared and cell type-specific functional pathways that are implicated in aging and neurodegeneration. Concomitantly, Alzheimer's disease (AD)-associated transcriptomic signature in microglia that arises from aging is reduced. These results show the feasibility of reversing brain aging by pharmacological means, provide mechanistic insights into the neurological benefits of GLP-1RAs, and imply that GLP-1R agonism may be a generally applicable pharmacological intervention for patients at risk of age-related neurodegeneration.

[1] Division of Neurology, Department of Medicine and Therapeutics, Faculty of Medicine, The Chinese University of Hong Kong, Shatin, Hong Kong. [2] Li Ka Shing Institute of Health Sciences, Faculty of Medicine, The Chinese University of Hong Kong, Shatin, Hong Kong. [3] Department of Chemical Pathology, Faculty of Medicine, The Chinese University of Hong Kong, Shatin, Hong Kong. [4] Department of Psychiatry, Faculty of Medicine, The Chinese University of Hong Kong, Shatin, Hong Kong. [5] Margaret K. L. Cheung Research Centre for Management of Parkinsonism, Faculty of Medicine, The Chinese University of Hong Kong, Shatin, Hong Kong. [6] Gerald Choa Neuroscience Centre, Faculty of Medicine, The Chinese University of Hong Kong, Shatin, Hong Kong. [7] Chow Yuk Ho Technology Centre for Innovative Medicine, Faculty of Medicine, The Chinese University of Hong Kong, Shatin, Hong Kong. [8] School of Biomedical Sciences, Faculty of Medicine, The Chinese University of Hong Kong, Shatin, Hong Kong. [9] Peter Hung Pain Research Institute, Faculty of Medicine, The Chinese University of Hong Kong, Shatin, Hong Kong. [10] These authors contributed equally: Zhongqi Li, Xinyi Chen, Joaquim S. L. Vong, Lei Zhao. ✉email: vctmok@cuhk.edu.hk; ho.ko@cuhk.edu.hk

Aging has long been considered irreversible. Nearly all cellular processes are implicated in or impacted by aging, ranging from metabolism, stress response, immune responses, cellular senescence, to gene expression, and genomic stability[1]. These complex molecular changes presumably lead to an alteration of cellular states and compositions in body organs[1,2], manifesting as age-related functional decline. Given the complexity of biological changes involved and a lack of easily targetable sets of driving pathways, anti-aging pharmacotherapy is considered highly challenging. However, it remains an attractive pursuit for tackling age-related disorders, such as neurodegeneration for which aging is the strongest risk factor. Slowing down or even reversing transcriptomic and functional alterations in the aging brain may provide a strategy for the primary prevention and even the treatment of neurodegenerative diseases.

Glucagon-like peptide-1 (GLP-1) is a peptide hormone produced peripherally by the intestinal L-cells for potentiating glucose-dependent insulin release, and centrally in the brain by preproglucagon neurons in the nucleus tractus solitarii[3]. In the past decade, multiple pharmacokinetically optimized GLP-1 receptor (GLP-1R) agonists (GLP-1RAs) have been approved for the clinical treatment of diabetes mellitus. Remarkably, recent clinical studies provided compelling evidence that GLP-1RAs exhibit neuroprotective effects beyond that conferred by glycemic control, reducing the incidences of cognitive decline and Parkinson's disease (PD) in diabetic patients[4,5]. Additionally, GLP-1RAs may slow the progression of established Alzheimer's disease (AD) and PD in non-diabetic patients[6,7]. Mechanistically, apart from alleviating neuroinflammation in animal models of neurodegeneration[8,9], we recently demonstrated that exenatide (a GLP-1RA) treatment partially reverses age-related transcriptomic changes in brain endothelial cells (ECs) and reduces nonspecific blood-brain barrier (BBB) leakage[10]. How GLP-1RA treatment impacts glial and other neurovascular cell types, whose age-related expression changes also play crucial roles in brain aging and degeneration, remained unclear.

## Results and discussion
### Systemic GLP-1RA treatment reverses aging-associated transcriptomic changes in diverse mouse brain cell types.
We hypothesized that GLP-1RAs may be potent anti-aging therapeutics that impact diverse brain cell types. We thus performed single-cell transcriptomic profiling experiments in young adult (2–3 months old (m.o.)), aged (18–20 m.o.) and exenatide-treated aged (18–20 m.o. with prior daily intraperitoneal injection for ~1 month, see "Methods" section) C57BL/6 mice (Fig. 1a, b and Supplementary Fig. 1), to examine the genome-wide expression changes in glial and neurovascular cells in aging, and their modulation by GLP-1R agonism. We calculated significant differentially expressed genes (DEGs) (defined as those with false discovery rate (FDR)-adjusted $P$-value < 0.05) for each cell type, to analyse their patterns of expression changes in aging and the effects of exenatide treatment. We obtained DEGs with similar magnitudes of change as previous studies[1,2], which were generally not large (Fig. 1c), likely due to comparisons being made between normal aging and young adulthood. At 18–20 m.o., however, these expression changes already have age-related functional change correlates in the C57BL/6 mice, including those that impact cognitive[11–14], glial (recently reviewed by ref. [15]), and neurovascular functions[10,16].

In line with our previous report[10], the aging-associated transcriptomic changes in brain ECs were partially reversed by exenatide treatment (Fig. 1c, d). Strikingly, the transcriptomic reversal effect of the GLP-1RA was even more profound in other

brain cell types (Fig. 1c, d). These included multiple glial (Fig. 1c, d, AC: astrocyte; OPC: oligodendrocyte precursor cell; MG: microglia) and mural (Fig. 1c, d, SMC: smooth muscle cell; PC: pericyte) cell types. This phenomenon was also observed, although weaker, in oligodendrocyte (OLG) and perivascular macrophage (MAC) (Fig. 1c, d). Overall, the reversal effect on glial cell age-related transcriptomic changes appeared to be the most prominent in AC and OPC, followed by MG (Fig. 1d). Among vascular cells, the effects were stronger in SMC and PC compared with that in EC (Fig. 1d). Further supporting the aging reversal effects, in seven out of the eight cell types analyzed (i.e., all except OLG), the number of significant DEGs for exenatide-treated aged vs. young adult group comparison was much smaller than that for untreated aged vs. young adult group comparison (Supplementary Fig. 2).

### Functional associations of the aging reversal effects at the transcriptomic level.
We next asked if the expression changes reversed by GLP-1RA treatment were functionally relevant, and which cellular pathways may be affected. Pathway enrichment analysis for each cell type on their most prominent reversed DEGs highlighted the amelioration of age-related expression changes involved in extensive cellular functions (Fig. 2a). In most cell types, these included genes mediating glucose/energy, lipid, and protein metabolic processes (Fig. 2a), as well as transcriptional and translational regulation (Fig. 2a). We also noted cell type-specific changes by pathway analysis and examining the expression changes of selected genes with significant functional roles. In aged brain ACs, upregulation of immune response-related genes and downregulation of homeostatic function-related genes occur (Fig. 2b) (also see ref. [17]). In our dataset, the ACs from the exenatide-treated aged mice appeared to partially revert to a younger phenotype, with downregulation of several complement component 1q genes (Fig. 2b), and upregulation of subsets of genes encoding synaptic modification-related proteins, metabolite receptors and transporters, neurotransmitter receptors and ion channels (Fig. 2b).

Immune response and cytokine signaling-related genes were especially prominent among DEGs reversed in MG and MAC, followed by EC (Fig. 2a). Indeed, previous studies reported that MGs in the aging and AD brain exhibit pro-inflammatory phenotypes[1,18–20]. A subset of MGs in AD mouse models are characterized by a disease-associated microglia (DAM) state, whose transcriptomic signature also increases in aging[1,20]. After exenatide treatment, apart from the reversed expression changes of several microglial activation-associated genes similar to what we previously reported[10] (Fig. 2b), we noted that the MGs also showed an upregulation of multiple homeostatic function-related and immune response inhibitory genes (Fig. 2b), and decreased AD-associated MG signature[1,20] scores (Fig. 2c, d). In SMC, we found post-treatment reversal of expression changes involving important functional processes associated with age-related vascular stiffening, such as calcium signaling, extracellular matrix (ECM) remodelling and contractile pathways (Fig. 2a, b).

### Consistency of the transcriptomic aging reversal effects across cell subtypes and datasets.
As we found prominent transcriptomic reversal effects in ACs (Fig. 1c, d), whose molecular phenotypes vary across brain regions[21], we asked if the age-related transcriptomic alterations and the GLP-1RA treatment effect in ACs may exhibit regional specificity. We identified four main regional AC subtype clusters, namely telencephalic AC clusters 1 and 2 (ACTE1 and ACTE2, respectively), and non-telencephalic AC clusters 1 and 2 (ACNT1 and ACNT2, respectively), using the expression patterns of known regional AC

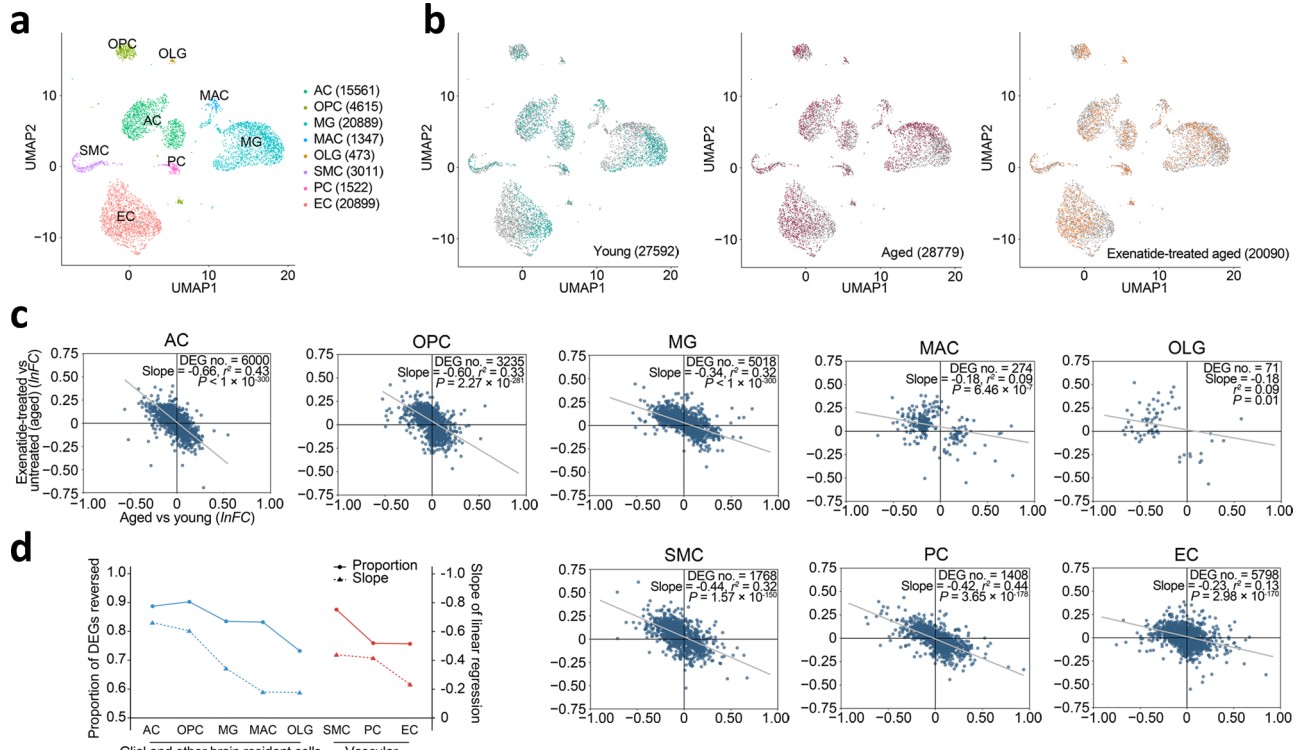

**Fig. 1 Reversal of glial and neurovascular cell transcriptomic aging signatures by GLP-1RA treatment. a** UMAP visualization of the major cell type clusters identified and analysed. Numbers in brackets: cell numbers for the respective cell types. AC astrocyte, OPC oligodendrocyte precursor cell, MG microglia, MAC perivascular macrophage, OLG oligodendrocyte, SMC smooth muscle cell, PC pericyte, EC endothelial cell. **b** UMAP visualization of the single-cell transcriptomes from young adult, aged, and GLP-1RA (exenatide)-treated aged mouse brains. For each plot, colored dots highlight cells from the respective labeled group, while gray dots are cells from the other groups. Numbers in brackets: cell numbers for the respective groups in the dataset ($n = 3$ animals for each group). For clarity, 6000 cells were subsampled for visualization in each plot in **a** and **b**. **c** Age-related expression changes (x-axis) plotted against post-exenatide treatment expression changes (y-axis) in glial (AC, OPC, MG, and OLG), vascular (EC, PC, and SMC) cell types, and MAC. Each dot represents one differentially expressed gene (DEG). lnFC: natural log of fold change. Gray lines: lines of best fit by linear regression. **d** Proportions of DEGs reversed and the slopes of lines of best fit by linear regression shown in **c** in the different cell types. See Supplementary Data 1 for source data underlying (**c**).

marker genes[21] (Supplementary Fig. 3a, b). Despite differences in their molecular characteristics (Supplementary Fig. 3a, b), the AC subtypes shared highly consistent age-related differential expressions (Supplementary Fig. 4a). The expression changes induced by exenatide treatment were generally even more consistent in the AC subtypes (Supplementary Fig. 4a). Consequently, the transcriptomic reversal effect of exenatide on the aging-associated expression changes was well conserved across all four regional AC subtypes examined (Supplementary Fig. 4b). We also performed similar analysis on the mural cell subtypes that segregate along the arteriovenous axis[22] (Supplementary Fig. 5a, b). Pairwise comparison of the differential expressions likewise revealed shared age-related expression changes and reversal by GLP-1RA treatment in all four mural cell subtypes (Supplementary Fig. 6a, b), despite their intrinsic transcriptomic differences (Supplementary Fig. 5a, b).

Finally, to test the robustness of the results, we verified that most of the top reversed age-related DEGs (i.e., those included in the pathway enrichment analysis) in EC, MG, and SMC had opposite directions of change in aging and after GLP-1RA treatment, with additional experiment comparing exenatide-treated aged mice to their vehicle-treated counterparts (Supplementary Fig. 7a, b). In addition, we also obtained qualitatively similar results on the genome-wide reversal of aging-associated differential expressions for several major brain cell types, including AC, OPC, MG, MAC, SMC, and EC (Supplementary Fig. 7c), based on an independent dataset from our prior study[10].

## Conclusions

In conclusion, we demonstrated that a generalized reversal of functionally relevant transcriptomic changes at the genome-wide level in multiple glial and vascular cell types in the aged brain is pharmacologically achievable with GLP-1R agonism. Further studies are required to assay the molecular and functional changes in neuronal circuits and glial cells with GLP-1R agonism in the aged brain. For example, while DAM is associated with neuropathology development in an AD mouse model[20], it remains incompletely understood what the functional significance of DAM-like MGs are in the aged brain. It has been shown that microglial depletion and repopulation in the aged mouse brain confers cognitive benefits[23]. Based on our results, we propose that the DAM-like MGs may be age-primed MGs with weakened homeostatic functions that are partially restored by GLP-1RA treatment—a hypothesis to be further tested.

In all, we speculate that the profound pleiotropic effects of GLP-1RA on brain aging may depend on a combination of central and peripheral mechanisms. Centrally in the brain, subsets of GLP-1R-expressing neurons and glial cells may be directly targeted by GLP-1RA[9,24]. Peripherally, GLP-1RA-mediated improvement in metabolic profiles or immunomodulation may impact the glial and neurovascular cells, as they are responsive to compositions of the circulation[25] and interact with peripheral immune cells[26]. Clarifying these possibilities will require systematic studies involving knockout or knockdown of GLP-1R in the respective putative target cell types (e.g., neuron, MG). The knowledge will also instruct further developments of

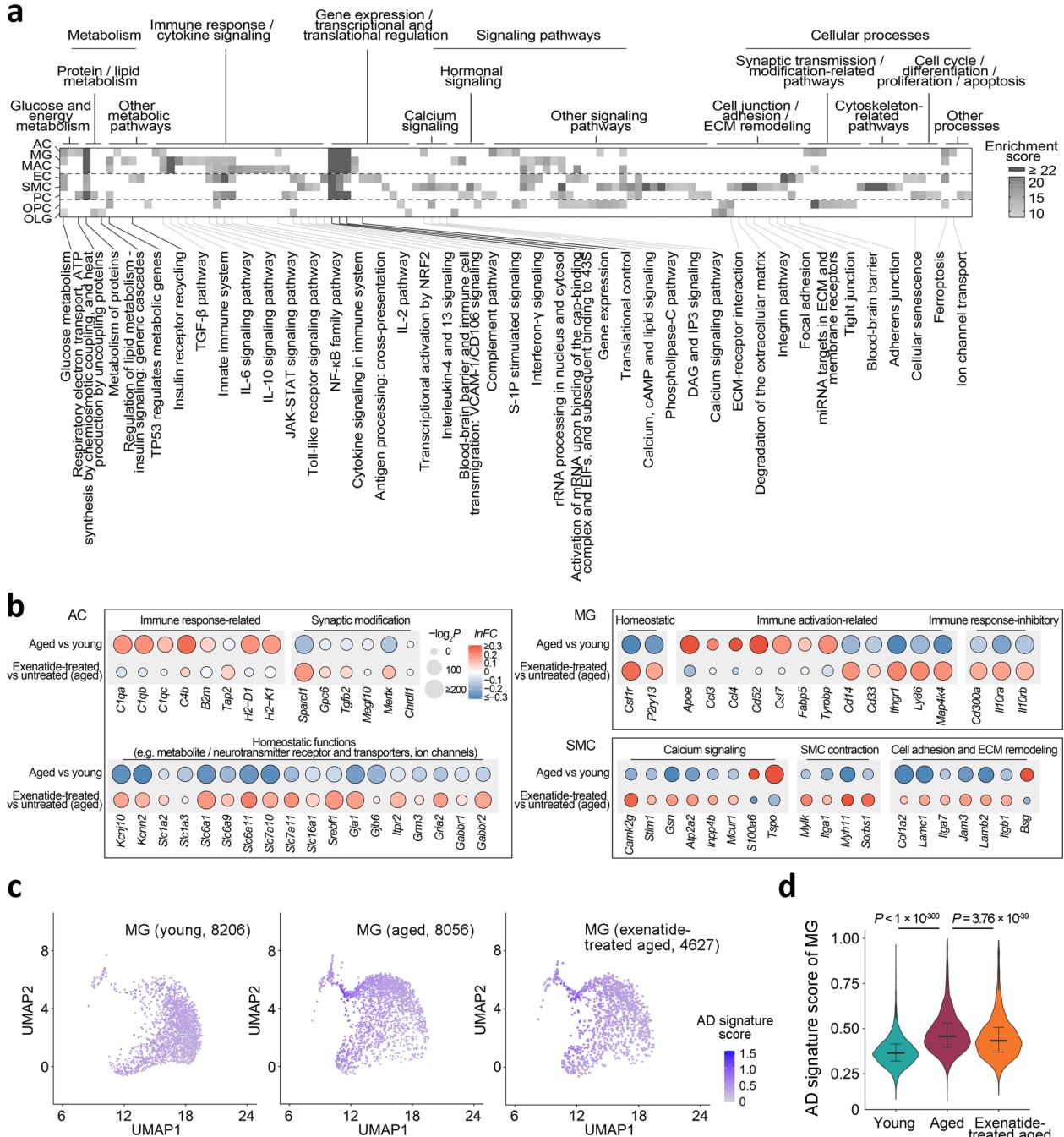

**Fig. 2 Functional pathway analysis of the GLP-1RA treatment-reversed age-related transcriptomic changes. a** Pathways with significant enrichment among the most prominent age-related expression changes reversed by exenatide treatment in the different cell types. **b** Dot plots of expression changes of selected functionally important genes in AC, MG, and SMC, in aging and post-exenatide treatment. **c** UMAP visualization and **d** violin plots of AD signature scores in the MGs from young adult, aged and exenatide-treated aged mouse brains. Numbers in brackets in **c**: cell numbers for the respective groups in the dataset. Horizontal lines in **d** represent medians (young adult, 0.36; aged, 0.46; exenatide-treated aged: 0.43), 25th and 75th percentiles. Comparison of Alzheimer's disease (AD) signature scores in the MGs among the three groups: $P = 2.2 \times 10^{-16}$, Kruskal–Wallis test; the $P$-values of pairwise comparisons by post hoc Dunn's test are shown in **d**. Cell type abbreviations: same as in Fig. 1. See Supplementary Data 1 for source data underlying **a** and **b**.

small-molecule GLP-1RAs[27–29], which may need to cross the BBB to act on the cell type target(s) necessary to attain any centrally mediated anti-brain aging effects.

## Methods

**Animal subjects.** All experimental procedures were approved in advance by the Animal Research Ethical Committee of CUHK, and were carried out in accordance

with the Guide for the Care and Use of Laboratory Animals. C57BL/6J mice were provided by the Laboratory Animal Service Center of CUHK and maintained at controlled temperature (22–23 °C) with an alternating 12-h light/dark cycle with free access to standard mouse diet and water. The ambient humidity was maintained at <70 % relative humidity. Male mice of two age groups (2–3 months old and 18–20 months old) were used for experiments. For the treatment groups, exenatide (5 nmol/kg bw, Byetta, AstraZeneca LP) or saline vehicle (0.9% w/v sodium chloride) was intraperitoneally (I.P.) administered (volume: 250 µl per 30 g

bw) daily starting at 17–18 months old for 4–5 weeks prior to transcriptomic profiling experiments.

**Single-cell dissociation and RNA sequencing**. Results reported in the study consisted of three datasets. The main results presented in all figures except Supplementary Fig. 7a, c were based on a dataset with young adult, aged and exenatide-treated aged mouse groups with cDNA library constructed using the 10× Chromium Single Cell 3′ Reagent Kits v3 (10× Genomics, USA) (hereafter referred to as the v3 kit-based dataset) (three mice per experimental group, one experimental batch). The second dataset with exenatide-treated and vehicle-treated aged mouse groups (hereafter referred to as the vehicle-controlled dataset) presented in Supplementary Fig. 7a, b was based on the 10× Chromium Single Cell 3′ Reagent Kits v2 (10× Genomics, USA) (three mice per experimental group, 1 experimental batch). The third dataset presented in Supplementary Fig. 7c was reported in our previous study[10] (with each batch consisting of young adult, aged, and exenatide-treated aged mouse groups, five mice per experimental group, three experimental batches, hereafter referred to as the v2 kit-based dataset).

For the v3 kit-based and vehicle-controlled datasets, identical brain tissue digestion, cell dissociation and single-cell RNA sequencing protocols were adopted as our previous study[10]. Mice were deeply anaesthetized and perfused transcardially with 20 ml of ice-cold phosphate buffered saline (PBS). Mice were then rapidly decapitated, and whole brains (except cerebellum) were immersed in ice-cold Dulbecco's modified Eagle's medium (DMEM, Thermo Fisher Scientific, USA). The brain tissues were cut into small pieces and dissociated into single cells using a modified version of the Neural Tissue Dissociation Kit (P) (130-092-628, Miltenyi Biotec, USA). Myelin debris was removed using the Myelin Removal kit II (130-096-733, Miltenyi Biotec, USA). Cell clumps were removed by serial filtration through pre-wetted 70-μm (#352350, Falcon, USA) and 40-μm (#352340, Falcon, USA) nylon cell strainers. Centrifugation was performed at 300×*g* for 5 min at 4 °C. The final cell pellets were resuspended in 500–1000 μl FACS buffer (DMEM without phenol red (Thermo Fisher Scientific, USA), supplemented with 2% fetal bovine serum (Thermo Fisher Scientific, USA)). For the v3 kit-based dataset, the single-cell suspension directly proceeded to cDNA library construction, while for the vehicle-controlled dataset the single-cell suspension was kept at −80 °C overnight and then thawed for cDNA library construction.

Single-cell RNA sequencing libraries were generated using the Chromium Single Cell 3′ Reagent Kit v3 (for the v3 kit-based dataset) or v2 (for the vehicle-controlled dataset) (10× Genomics, USA). Briefly, single-cell suspension at a density of 500–1000 cells/μL in FACS buffer was added to real-time polymerase chain reaction (RT-PCR) master mix aiming for sampling of up to 8000 cells, and then loaded together with Single Cell 3′ gel beads and partitioning oil into a Single Cell 3′ Chip (10× Genomics, USA). RNA transcripts from single cells were uniquely barcoded and reverse-transcribed within droplets. cDNA molecules were preamplified and pooled, followed by library construction. All libraries were quantified by Qubit and RT-PCR on a LightCycler 96 System (Roche Life Science, Germany). The size profiles of the pre-amplified cDNA and sequencing libraries were examined by the Agilent High Sensitivity D5000 and High Sensitivity D1000 ScreenTape Systems (Agilent, USA), respectively. All single-cell libraries were sequenced with a customized paired-end with single indexing (26/8/98-bp) format. All single-cell libraries were sequenced on a NextSeq 500 system (Illumina, USA) using the NextSeq 500 High Output v2 Kit (Illumina, USA). The data were aligned in Cell Ranger (v3.0.0, 10× Genomics, USA). For the v3 kit-based dataset, the library sequencing saturation was on average 68.73%. Compared to the v2 kit-based dataset in our previous study[10], the v3 kit-based dataset had much improved mRNA capture efficiency (median genes per cell: 2091, range: 505–5791), allowing more sensitive detection of differentially expressed genes (DEGs). For the vehicle-controlled dataset, the sequencing saturation was on average 84.1% (median genes per cell: 1229, range: 1175–1283).

**Data quality control, single-cell transcriptome clustering, and visualization**. Data processing and visualization were performed using the Seurat package (v3.1.5)[30] and custom scripts in R (v3.6.1). The raw count matrix was generated by default parameters (with the mm10 reference genome). For the v3 kit-based dataset, there were 106,902 cells in the primary count matrix. Genes expressed by fewer than five cells were removed, leaving 21,259 genes in total. Among these genes, 3000 high-variance genes were identified by the Seurat *FindVariableFeatures* function. The dataset was filtered to exclude low-quality cells by the following criteria: (1) <5% or >95% UMI count or gene count, or (2) proportion of mitochondrial genes >20%. Gene count normalization and high-variance gene identification were applied to the raw data of 78,490 cells retained for further analysis. The *SCTransform* function in the Seurat package was used for expression normalization by fitting the data to a negative binomial regression model. For dimensionality reduction, principal component analysis (PCA) was applied to compute the first 30 top principal components. Clustering was carried out by the Seurat functions *FindNeighbors* and *FindClusters*. Modularity optimization was then performed on the shared nearest neighbor graph results from *FindNeighbors* for clustering (resolution parameter: 1.4). We employed Uniform Manifold Approximation and Projection (UMAP)[31], a manifold learning-based technique for dimensionality reduction, for the visualization of single-cell transcriptomes and clustering results. For the vehicle-controlled dataset, there were 6703 cells in the primary count matrix. Genes expressed by fewer than three cells were removed,

leaving 16,684 genes in total. Subsequent data processing was identical to that performed for the v3 kit-based dataset.

**Cell type and subtype identification**. To identify primary cell types, we employed known cell type-specific marker genes and examined their expression levels among all initial clusters included. We excluded clusters with high expression of two or more cell type-specific marker genes. These included clusters with high expression of both endothelial cell and pericyte markers in the v3 kit-based and the vehicle-controlled datasets, corresponding to contamination of pericytes by endothelial cell fragments[22,32,33]. For the v3 kit-based dataset, the remaining clusters were classified into primary cell types, whereby eight were analysed in this study (see Fig. 1 and Supplementary Fig. 1). Telencephalic and non-telencephalic AC subtypes were identified by unsupervised clustering of the AC transcriptomes (after expression normalization among ACs), followed by examining the expression patterns of regionally specific marker genes[21] (see Supplementary Fig. 3). We did not analyze a relatively small number of midbrain astrocytes (i.e., small sample size compared to telencephalic and non-telencephalic astrocytes, whereby midbrain astrocytes constituted <1% of the total number of astrocytes sampled). To identify SMC subtypes, the *CellAssign* algorithm was employed[34], using previously reported SMC subtype marker genes[22] as prior (see Supplementary Fig. 5). For the vehicle-controlled dataset, three major cell types (EC, MG, and SMC) with sufficient cell numbers underwent further differential expression analysis.

**Differential expression analysis**. The Seurat *FindMarkers* function and the MAST package (v1.8.2) were employed for the calculation of differentially expressed genes (DEGs) with associated false discovery rate (FDR)-adjusted *P*-value and magnitude of change expressed in natural log of fold change (*lnFC*) for each cell type or subtype. We define significant DEGs as those with FDR-adjusted $P$-value < 0.05.

**Transcriptomic reversal effect, pathway enrichment, and consistency of differential expressions analyses**. For each cell type (or subtype) in the v3 kit-based or the v2 kit-based dataset[10], the union of significant DEGs in either or both the aged versus young adult group, and the exenatide-treated versus untreated aged group comparisons were included for transcriptomic reversal analysis. Proportions of genes with opposite directionality of changes (i.e., reversed), linear regression analysis with slope of line of best fit, associated $r^2$ and $P$-values were obtained.

For pathway enrichment analysis of the reversed DEGs of a given cell type using the v3 kit-based dataset, DEGs with fold changes ranking within top 500 in both age-stratified and treatment-stratified comparisons and opposite directionality of changes were included (i.e., top reversed DEGs). Pathways with significant enrichment were identified using the GeneAnalytics platform[35]. To examine the consistency of differential expressions across cell subtypes in the v3 kit-based dataset, for each pair of cell subtypes, the union of their significant DEGs in a given comparison (i.e., age-stratified or treatment-stratified) were included. Linear regression analysis was then carried out with slope of line of best fit, associated $r^2$ and $P$-values obtained.

For the vehicle-controlled dataset, differential expressions with the associated *lnFC* and FDR-adjusted $P$-values were calculated for the top reversed DEGs (defined above), and plotted against the *lnFC* from the aged vs. young adult mouse group comparison in the v3 kit-based dataset.

**Calculation of Alzheimer's disease (AD)-associated microglia signature scores**. For each single MG transcriptome, the AD-associated MG signature score was calculated using the *AddModuleScore* function in the Seurat package, as the average normalized expression levels of top 100 upregulated genes in AD-associated MGs[20] subtracted by the aggregated expression of control feature sets[1].

**Statistics and reproducibility**. For the v3 kit-based dataset, we used three mice per group (young adult, aged and exenatide-treated aged groups) for the scRNA-seq experiments. For the vehicle-controlled dataset, we used three mice per group (vehicle-treated and exenatide-treated aged groups) for the scRNA-seq experiments. The statistical tests were performed in R (v3.6.1) as indicated at the respective places in the main text. We confirmed reproducibility of the main findings with independent batches of experiments and datasets (see Supplementary Fig. 7).

**Reporting summary**. Further information on research design is available in the Nature Research Reporting Summary linked to this article.

## Data availability

Source data underlying Figs. 1–2 are presented in Supplementary Data 1. The single-cell RNA sequencing data presented in the main figures has been deposited at the Broad Institute Single Cell Portal and are accessible at the following URL: [https://singlecell.broadinstitute.org/single_cell/study/SCP1182/glp1ra-brain-aging-reversal]. The dataset reported in our previous study (Zhao et al.) is accessible at the following URL: [https://singlecell.broadinstitute.org/single_cell/study/SCP829/aging-mouse-brain-kolab]. All other data are available from the corresponding authors upon reasonable request.

## Code availability

The R code for data analysis is available at GitHub at the following URL: [https://github.com/RichardLZQ/NVUB5_code].

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

## Acknowledgements

We thank Dennis Lo and Rossa Chiu for generous support; Becky Yung, Florence Yau, Anki Miu, and Rebecca Chau for administrative support to the project. This work was funded by a Croucher Innovation Award (CIA20CU01) from the Croucher Foundation (H.K.); Faculty Innovation Awards (FIA2020/B/01 and FIA2017/B/01) from the Faculty of Medicine, CUHK (H.M.L. and H.K.); the Collaborative Research Fund (C6027-19GF) and the Area of Excellence Scheme (AoE/M-604/16) of the University Grants Committee of Hong Kong (H.K.).

## Author contributions

Z.L., X.C., J.S.L.V., and L.Z. carried out the in vivo treatment and single-cell RNA sequencing experiments. L.Y.C.Y. assisted the experiments. Z.L. and X.C. analysed the data. J.H., B.I., Y.K.W., and H.M.L. contributed to data interpretation. Z.L., X.C., J.H., H.M.L., V.C.T.M., and H.K. wrote the paper with input from all authors. All authors read and approved the final manuscript.

## Competing interests

C.U.H.K. filed a patent application based partly on the discovery reported in the study. Z.L., X.C., J.S.L.V., L.Z., J.H., L.Y.C.Y., H.M.L., V.C.T.M., and H.K. are inventors on the patent. Exenatide used in the study was provided by AstraZeneca Hong Kong Limited. AstraZeneca Hong Kong Limited had otherwise no role in the funding, design, execution or data interpretation of the study. All other authors declare no competing interests.
