## [Peer Review File · Communications Biology]

Reviewers' comments:

Reviewer #1 (Remarks to the Author):

Thanks for asking me to review this paper. The authors present the data from single cell transcriptomic analysis of multiple glial and neurovascular cell types comparing young and aged cells, as well as exenatide treated and untreated aged cells. They find differentially expressed genes with aging and analogous gene expression changes with exenatide treatment. Up to 90% of the differentially expressed genes with aging are reversed with exenatide. These results are additionally compelling given the clinical data emerging that GLP-1 agonism may have beneficial effects on neurodegeneration.

I have only a few minor comments;

1. The UMAP methods are referenced to an Arxiv paper which has already been cited 1747 times. It would be useful to include a brief introductory sentence to the text to save the uninitiated reader from having to look up the UMAP methods.
2. There are no neuronal data- presumably these will be subject of a future publication. Can the authors please clarify if this is the case.

Reviewer #2 (Remarks to the Author):

This Brief Communication by Zhongqi Li et al, 'Genome-wide reversal of glial and neurovascular cell transcriptomic aging signatures by GLP-1R agonism' describes a preliminary data set that includes three groups of mice that are young (2-3 months), old (18 – 20 months), or old and treated with exenatide, a GLP-1R agonist that was intraperitoneally (I.P.) administered daily for 4-5 weeks. Single cell RNAseq was used to compare these three groups, or rather the difference between differentially expressed genes (DEGs) in aged vs young and aged vs treated group.

These data are certainly interesting, as options to increase 'healthy aging' would be a significant step forward. However, I have certain questions on the experimental design and readouts. Firstly, do the authors think that a maximum 1 +/- 0.35 fold change ($\ln[0.3]$) in gene expression that is shown for the markers they have selected is significant enough in this context?

I agree that when entire signalling pathways are changed in similar direction, it can provide confidence that even very small changes such as these may well have overall relevance. However, the data are somewhat weakened by the fact that the aged animals are 'untreated' rather than given a vehicle injection. 4-5 weeks of daily i.p injections is a significant undertaking for an animal. How can the authors be sure the effects are not due to daily stress of restraint/injection etc?

Also, were all the groups taken at the same time? Was the single cell extraction performed for all three groups on the same day? Batch effects could easily account for changes such as those described.

The authors talk about 'reversal' of aging signatures, rather than 'healthy aging'. To better support this concept, a direct comparison between young and aged exenatide treated animals may also be instructive. As a 'reversal' would show far fewer DEGs between the groups. Though this comparison would suffer the same issue that the young animals were not also injected daily for 5 weeks.

A large part of Figure 2 is dedicated to the microglia signature in aging. How are these data different to what this same group published in the supplemental figures in their Nature Communications paper in 2020?

For Figure 2d. The authors report a significant reduction on 'AD signature scores of microglia'. As with the much of the presented data, the p values are extremely small, yet the fold

change/difference in means is negligible.

Do the authors think that the 'normalized expression levels of top 100 upregulated genes in AD-associated MGs subtracted by the aggregated expression of control feature sets' resulting in a mean expression of approx. 4.2 and aged animals to 4.0 in treated animals means anything biologically? Do the authors have any evidence that microglia are functionally changed with treatment of exenatide?

Rebuttal Letter

We thank both reviewers for the important comments that helped us significantly improve the manuscript. We have now revised the paper to address the concerns raised, and implement the changes suggested.

Major changes include the following:

1. Inclusion of results from an additional dataset with vehicle-controlled aged group for comparison against GLP-1RA-treated aged group, to verify that we can confidently attribute the observed therapeutic effects to GLP-1RA treatment (new Supplementary Fig. 7a, b).
2. Additional analysis from an independent dataset from our previous study (Zhao *et al.*, 2020) to further show the robustness of the major findings (new Supplementary Fig. 7c).
3. Additional analysis to show that post-GLP-1RA treatment, we indeed obtained far fewer significant DEGs for the different cell types in the aged mouse brain (new Supplementary Fig. 2).
4. Revision of the writing with additional discussions to better reflect both the significance and potential limitations of the findings.
5. Revision of the Methods section for the inclusion of additional datasets and analysis, and to ensure clarity of presentation.

All revised texts are highlighted in red and underlined in the manuscript. Please see below for detailed point-by-point responses to each of the comments and suggestions.

Reviewer #1 (Remarks to the Author):

Thanks for asking me to review this paper. The authors present the data from single cell transcriptomic analysis of multiple glial and neurovascular cell types comparing young and aged cells, as well as exenatide treated and untreated aged cells. They find differentially expressed genes with aging and analogous gene expression changes with exenatide treatment. Up to 90% of the differentially expressed genes with aging are reversed with exenatide. These results are additionally compelling given the clinical data emerging that GLP-1 agonism may have beneficial effects on neurodegeneration.

We thank the reviewer for the positive remarks.

I have only a few minor comments;

1. The UMAP methods are referenced to an Arxiv paper which has already been cited 1747 times. It would be useful to include a brief introductory sentence to the text to save the uninitiated reader from having to look up the UMAP methods.

Thanks for the suggestion. We have now revised the manuscript as advised (page 8, paragraph 1), where the corresponding sentence is now: “*We employed Uniform Manifold Approximation and Projection (UMAP), a manifold learning-based technique*

for dimensionality reduction, for the visualization of single-cell transcriptomes and clustering results”.

2. There are no neuronal data- presumably these will be subject of a future publication. Can the authors please clarify if this is the case.

Thanks for bringing up this interesting point. Just as the reviewer speculated, we are currently working on a follow-up project on the mechanistic aspects, and also plan to include neuronal data in a future publication. In the revised manuscript, to relate to this point we now mention that “Additionally, further studies are required to assay the molecular and functional changes in neuronal circuits and glial cells with GLP-1R agonism in the aged brain” (page 5, paragraph 2).

Reviewer #2 (Remarks to the Author):

This Brief Communication by Zhongqi Li et al, ‘Genome-wide reversal of glial and neurovascular cell transcriptomic aging signatures by GLP-1R agonism’ describes a preliminary data set that includes three groups of mice that are young (2-3 months), old (18 – 20 months), or old and treated with exenatide, a GLP-1R agonist that was intraperitoneally (I.P.) administered daily for 4-5 weeks. Single cell RNAseq was used to compare these three groups, or rather the difference between differentially expressed genes (DEGs) in aged vs young and aged vs treated group.

These data are certainly interesting, as options to increase ‘healthy aging’ would be a significant step forward. However, I have certain questions on the experimental design and readouts. Firstly, do the authors think that a maximum 1 +/- 0.35 fold change ($\ln[0.3]$) in gene expression that is shown for the markers they have selected is significant enough in this context?

We thank the reviewer for the comment that our data is interesting, as well as remarking that having more therapeutic options to promote healthy aging is significant.

We agree with the reviewer that for a significant proportion of the genes, the fold-changes of age-related expression changes found were not large. We believe this was primarily due to the fact that comparison was made between young adult (2 – 3 months old) and normally aging C57BL/6 mice (18 – 20 months old), hence we would not expect large differential expression changes. The magnitude of expression changes at the transcriptional level we found were similar as those reported in other recent publications that performed similar quantifications in the aged mouse brain at similar or older ages (e.g., Ximerakis et al., *Nature Neuroscience*, 2019; the Tabula Muris Senis study, *Nature*, 2020).

Despite this, we propose that the expression changes we and others reported are still *potentially* functionally significant. We base our judgment on the existing literature reporting that at 18 – 20 months old (or younger), the mouse brain already exhibit functional changes impacting learning and memory (well reported in multiple studies, e.g., Verbitsky et al., *Learning and Memory*, 2004; C. de Fiebre et al., *Age*, 2006; Yang et

al., *Frontiers in Aging Neuroscience*, 2019; Hamieh et al., *Brain Research*, 2021), glial homeostatic functions (*c.f.* recently reviewed by Salas et al., *Neurobiology of Disease*, 2020) and neurovascular functions (e.g., Park et al., *Stroke* 2014; and Zhao et al., *Nature Communications*, 2020 (our previous publication)), that correlate with the molecular changes. We also agree with the reviewer that when numerous genes involved in a given signaling pathway are changed, the likelihood of the collective effects of expression changes being functionally significant is higher – hence we carried out pathway enrichment analysis using the top differentially expressed genes to infer the *potential* cellular functions impacted.

We now discuss the above-mentioned points (page 2, paragraph 3; page 3, paragraph 1) in the revised manuscript, which are indeed important considerations when interpreting the potential functional relevance of the expression changes reported.

I agree that when entire signalling pathways are changed in similar direction, it can provide confidence that even very small changes such as these may well have overall relevance. However, the data are somewhat weakened by the fact that the aged animals are ‘untreated’ rather than given a vehicle injection. 4-5 weeks of daily i.p injections is a significant undertaking for an animal. How can the authors be sure the effects are not due to daily stress of restraint/injection etc?

We thank the reviewer for reminding us of this important point. We in fact did carry out experiments in aged mice with exenatide treatment compared to vehicle treatment. In the revised manuscript, we now include these results as supplementary information (page 4, paragraph 3 and page 5, paragraph 1; Supplementary Fig. 7a, b).

In that batch of experiment (hereafter referred to as the vehicle-controlled dataset, which was based on the 10X Chromium Single Cell 3’ Reagent kit v2), we lost astrocytes (AC) (presumably due to sample freezing and thawing before library construction, a step other batches of experiments did not undergo) but obtained sufficient cell numbers for three major cell types, namely microglia (MG), smooth muscle cell (SMC) and endothelial cell (EC) (Supplementary Fig. 7a). Note that the dataset reported in the previous version of the paper was solely based on the 10X Chromium Single Cell 3’ Reagent kit v3 (hereafter referred to as the v3 kit-based dataset).

Despite that the vehicle-controlled dataset had limited cell numbers, we were able to use it to verify the main findings for MG, SMC and EC (Supplementary Fig. 7b). Specifically, we re-examined the top reversed differentially expressed genes (DEGs) from the v3 kit-based dataset, by plotting their expression changes in the vehicle-controlled dataset (*i.e.*, exenatide- vs vehicle-treated aged mice) against that in the aged vs young adult comparison from the v3 kit-based dataset (Supplementary Fig. 7b). Most of the DEGs, especially those that reached statistical significance in MG and EC in the vehicle-controlled dataset, exhibited the same reversal effect by exenatide with this analysis (Supplementary Fig. 7b, *left and middle panels*). In SMC, although most genes did not reach statistical significance individually in the vehicle-controlled

dataset, the expression reversal effect of exenatide treatment for most of the DEGs could still be verified (Supplementary Fig. 7b, *right panel*). We thus believe that our reported results with the v3 kit-based dataset are robust and reflect genuine GLP-1R agonist treatment effects.

Additionally, we were also able to reproduce qualitatively the same results based on 3 other independent batches of experiments (i.e., the dataset based on the 10X Chromium v2 kit reported in our previous study Zhao *et al.*, 2020), for numerous cell types including astrocyte (AC), oligodendrocyte precursor cell (OPC), MG, perivascular macrophage (MAC), SMC and EC (Supplementary Fig. 7c). We originally only presented results from the v3 kit-based dataset as it had better mRNA / transcript capture and gene detection sensitivity, and therefore higher data quality and DEG detection sensitivity. We however reckon that having additional verifications help to show the robustness of the results. In the revised manuscript, we thus include these results as supplementary information (page 4, paragraph 3 and page 5, paragraph 1; Supplementary Fig. 7c).

Also, were all the groups taken at the same time? Was the single cell extraction performed for all three groups on the same day? Batch effects could easily account for changes such as those described.

Yes, for the v3 kit-based dataset all three groups underwent brain dissection, tissue digestion and single cell isolation on the same day, while subsequent library preparation were also carried out together, hence the experimental data in the v3 kit-based dataset presented came from the same batch. We now clarify this point in the Methods section of the revised manuscript (page 6, paragraph 2).

In our previous study (Zhao *et al.*, 2020, using the 10X Chromium v2 kit), we underwent the same procedures for 3 batches of experiments (i.e., each batch had all three experimental groups). For that dataset, we indeed performed batch effect correction for single-cell transcriptome clustering, followed by differential expression analyses on the normalized expression counts as detailed in our previous publication. We obtained consistent results for several major cell types using that dataset (Supplementary Fig. 7c, also kindly see the responses above), thereby substantiating the robustness of the findings.

The authors talk about ‘reversal’ of aging signatures, rather than ‘healthy aging’. To better support this concept, a direct comparison between young and aged exenatide treated animals may also be instructive. As a ‘reversal’ would show far fewer DEGs between the groups. Though this comparison would suffer the same issue that the young animals were not also injected daily for 5 weeks.

We thank the reviewer for suggesting the analysis. We now present additional results (page 3, paragraph 1) on the number of significant DEGs (Supplementary Fig. 2) for aged *vs* young adult and exenatide-treated aged *vs* young adult comparisons. For the exenatide-treated aged *vs* young adult comparison, in 7 out of the 8 cell types analyzed we indeed obtained far fewer significant DEGs (i.e., all except OLG, see Supplementary

Fig. 2). We agree with the reviewer on the lack of a vehicle-treated young adult group as a limitation for this analysis, and yet believe that this would not affect our conclusions (kindly see responses to previous comment regarding vehicle control).

A large part of Figure 2 is dedicated to the microglia signature in aging. How are these data different to what this same group published in the supplemental figures in their Nature Communications paper in 2020?

We thank the reviewer for careful and detailed reading of our previous publication. The data presented in Fig. 2 of our current study are in fact complementary to that presented in our previous paper. In the current paper, we covered two aspects not presented in the previous publication, including (i) the upregulated transcription of immune response-inhibitory genes (Fig. 2b), and (ii) a more focused presentation of the signature genes of disease-associated microglia (DAM) that also become upregulated in aging (Fig. 2c, d). We considered different ways of presentation, yet prefer to keep the partially overlapping findings in Fig. 2b on microglial activation-associated genes as it is, for a more complete presentation together with Fig. 1 (which shows the genome-wide reversal), 2c and 2d. In the revised manuscript, we now refer to our previous publication to more explicitly describe which of the presented data in the current paper are new and complementary (page 4, paragraph 1): *“After exenatide treatment, apart from the reversed expression changes of several microglial activation-associated genes similar to what we previously reported (Fig. 2b), we noted that the MGs also showed an upregulation of multiple homeostatic function-related and immune response inhibitory genes (Fig. 2b), and decreased AD-associated MG signature scores (Fig. 2c, d).”*

For Figure 2d. The authors report a significant reduction on ‘AD signature scores of microglia’. As with the much of the presented data, the p values are extremely small, yet the fold change/difference in means is negligible. Do the authors think that the ‘normalized expression levels of top 100 upregulated genes in AD-associated MGs subtracted by the aggregated expression of control feature sets’ resulting in a mean expression of approx. 4.2 and aged animals to 4.0 in treated animals means anything biologically? Do the authors have any evidence that microglia are functionally changed with treatment of exenatide?

Thanks for this important comment regarding the magnitude of changes in AD signature scores in the aged brain MG after GLP-1RA treatment, in relation to the functional significance of DAM-like MG in the aged brain.

In the aged mouse brain, the median MG AD signature score was 0.46, compared to 0.36 in the young adult group (i.e., 0.10 higher in the aged brain MGs). Post-exenatide treatment, the median was reduced to 0.43 (i.e., 0.07 higher than young adult brain MGs), representing ~30% decrease in the age-related elevation in the median of signature score. We fully agree with the reviewer that the difference in any individual genes may be minimal and negligible. However, considering that the MG AD score was based on jointly analyzing the top 100 signature genes, we propose that the collective difference may not be negligible – as when the expression levels of 100 MG

genes altogether share more similarity with DAM in aging and such changes were reduced by ~30% on average for each gene after GLP-1RA treatment, the joint effects are more likely biologically significant.

On the other hand, the interpretation also depends on our understanding on the functional significance of the DAM-like MG in the first place. We acknowledge that while it has been demonstrated that the DAM may play a role in neuropathology development in the 5xFAD mouse model (Keren-Shaul et al., *Cell*, 2017), even with our findings it remains unclear how the functions of DAM-like MG in the aged brain are modulated after GLP-1RA treatment, nor do we know for sure what their functional significance in the aged brain are to begin with. Previously, it has been shown that depleting MGs in the aged mouse brain permits their repopulation, which confers cognitive benefits (Elmore et al., *Aging Cell*, 2018). We propose that the DAM-like MG may represent aged MG with weakened homeostatic functions and are primed to become reactive, via the downregulation of homeostatic function-related genes and altered expression of activation-associated genes (see Fig. 2d). In our previous study, we performed IBA1 staining, and noted a trend of decrease in MG density after exenatide treatment in the aged mouse brain. In this study, we additionally report the upregulation of both homeostatic function- and immune inhibitory function-related genes in the aged MG after GLP-1R agonist treatment, suggesting that GLP-1R agonism may partially reinstate the homeostatic functions of MG and reduce their priming. Certainly, these speculations will require future studies to validate. We now discuss these points in the revised manuscript (page 5, paragraph 2), to help readers better appreciate both the potential significance and the limitations of the findings.

REVIEWERS' COMMENTS:

Reviewer #1 (Remarks to the Author):

Thanks for asking me to look at this manuscript again. The authors have satisfactorily addressed my minor concerns.

Reviewer #2 (Remarks to the Author):

I would like to thank the authors for their careful revision of the manuscript and addressing some of the queries I had.

I would now fully endorse this manuscript for publication.

Sincerely,

Andy Greenhalgh